# Negative Regulatory Loop between Microphthalmia-Associated Transcription Factor (MITF) and Notch Signaling

**DOI:** 10.3390/ijms20030576

**Published:** 2019-01-29

**Authors:** Tamar Golan, Carmit Levy

**Affiliations:** Department of Human Genetics and Biochemistry, Sackler Faculty of Medicine, Tel Aviv University, Tel Aviv 69978, Israel; golantammy@gmail.com

**Keywords:** melanoma, notch signaling, MITF

## Abstract

Melanoma, a melanocyte-origin neoplasm, is a highly metastatic and treatment-resistance cancer. While it is well established that notch signaling activation promotes melanoma progression, little is known about the reciprocal interactions between Notch signaling and melanoma-specific pathways. Here we reveal a negative regulatory loop between Notch signaling and microphthalmia-associated transcription factor (MITF), the central regulator of melanoma progression and the driver of melanoma plasticity. We further demonstrate that Notch signaling activation, in addition to the known competition-based repression mechanism of MITF transcriptional activity, inhibits the transcription of MITF, leading to a decrease in MITF expression. We also found that MITF binds to the promoter of the gene encoding the master regulator of Notch signaling, recombination signal binding protein J kappa (RBPJK), leading to its upregulation. Our findings suggest that, once activated, Notch signaling represses MITF signaling to maintain the melanoma invasiveness and metastatic phenotype.

## 1. Introduction

Melanoma, a melanocytic neoplasm, is a highly lethal and treatment-refractory cancer [1,2,3,4] that affects an estimated 100,000 new patients worldwide each year [5]. Although new therapeutic approaches, such as immunotherapy, have improved prognoses, metastatic melanoma is resistant to most current therapies [5], and the estimated five year survival is a paltry 6% [5]. Clearly, novel approaches are required to improve this dire prognosis. 

One such potential new treatment strategy centers on Notch signaling [6,7]. This evolutionarily conserved pathway, which regulates a wide range of cellular processes, including cell proliferation, cell fate, and progenitor differentiation [8,9,10], plays an important role in melanoma. It does so by promoting tumor proliferation [11,12,13,14], invasion [11,13,15] and immunosuppression [6,14]. However, the reciprocal interactions between Notch signaling and melanocyte lineage-specific pathways have been understudied, information necessary for realizing the Notch pathway’s therapeutic potential. 

Notch signaling is activated by extracellular binding of a membrane-bound ligand expressed on the sender cell (Jagged1-2, Delta-like1, 3, or 4) to a single-pass transmembrane cell surface Notch receptor (Notch1-4) expressed on an adjacent receiver cell [8,16,17]. Ligand-receptor binding, which requires cell-cell contact, activates proteolytic cleavage of the Notch receptor by the gamma-secretase complex, which in turn causes the release of the Notch intracellular domain (NICD) [8,16,17]. NICD then translocates to the nucleus, where it interacts with recombination signal binding protein J kappa (RBPJK), converting it from a transcriptional repressor to an activator [18]. The NICD-RBPJK complex initiates the transcription of Notch target genes, including specific members of the Hes/Hey family [8,16,17,18].

The microphthalmia-associated transcription factor (MITF) is the melanocytic lineage master regulator [19]. It is a basic helix-loop-helix leucine zipper, dimeric transcription factor that can function as either a homo- or a heterodimer [19]. MITF plays a central role in melanoma progression [20,21] and is recognized as the major driver of melanoma plasticity [22,23,24]. Melanoma progression is thought to occur via a “phenotypic switch” model, in which cells continuously switch between proliferative and invasive phenotypes. In the early stage, cells express a proliferative signature gene set that allows proliferation at the primary lesion. As the disease progresses, microenvironmental cues trigger expression of the invasive gene set. These cells are able to migrate out of the primary site to a suitable distal site. Once at the metastatic location, the melanoma cells revert to the proliferative state, and a new cycle is started [23]. MITF is the key regulator of the melanoma phenotypic switch: It transcriptionally regulates the gene set responsible for both proliferative and invasive phenotypes. Cells that express high levels of MITF exist in a proliferative and weakly invasive state, whereas cells with low MITF levels are highly invasive and less proliferative [23,24,25,26]. 

At the mRNA level, there are at least 10 isoforms of MITF, resulting in eight distinct proteins that differ at their N-termini, and the expression patterns vary depending on the tissue [27]. Among the MITF isoforms are mouse-MITF (M-MITF), the master regulator of the melanocyte lineage [28], and heart-MITF (H-MITF), which is expressed in cardiomyocytes [29]. H-MITF plays a major role in pathological heart conditions, such as hypertrophy and heart failure [29], by regulating the expression of Erbin in the heart [30]. The Notch signaling pathway has a well-established role in multiple processes necessary for heart development, including cardiomyocyte differentiation, patterning of cardiac regions, and valve development [31]. Given the importance of both Notch and MITF signaling pathways in cardiac normal function and diseases, understanding the reciprocal regulation that exists between these two pathways will contribute to our understanding of pathological heart conditions and to the development of new therapeutic approaches. 

Our previous work showed that differentiated keratinocytes express Notch ligands in the melanoma microenvironment and demonstrated that the interaction of melanoma cells with keratinocytes activates Notch signaling in melanoma cells [15]. Activated Notch inhibits MITF function via a competition-based mechanism, thereby triggering the critical transit into the invasive melanoma stage [15]. We have also demonstrated that RBPJK occupies the promoters of MITF target genes and, by direct interaction with MITF, acts as its transcription co-factor [32]. In light of the major roles that impaired Notch signaling plays in melanoma progression and induction of metastatic ability [11,12,13,14,15] and in other diseases [10], we hypothesized that additional layers of regulation between Notch signaling and MITF signaling exist. Herein, we describe our discovery of a negative transcription-based regulatory loop between these two pathways. We demonstrate that Notch signaling inhibits MITF expression, whilst MITF increases the expression of RBPJK. Our findings suggest that activation of Notch signaling induces a negative regulatory loop that maintains low levels of MITF and, consequently, promotes a metastatic phenotype.

## 2. Results and Discussion

### 2.1. Notch signaling Decreases MITF Expression

We previously showed that Notch signaling alters the DNA binding capacity of MITF, thus inhibiting the transcriptional program mediated by MITF [15]. In contrast, Hes5, a known Notch signaling target gene [18], is upregulated upon MITF expression [15]. To further explore the regulatory interactions between Notch signaling and MITF, we took advantage of two types of melanoma cell line systems: Non-invasive melanoma cells (WM3682, WM3526) and cells with high invasive abilities (WM1716, WM3314) [28]. All four lines express detectable levels of the Notch receptor and are capable of Notch signaling activation upon interaction with a Notch ligand. However, the highly metastatic lines (WM1716, WM3314) have higher levels of Notch receptors and are characterized by constitutively active Notch signaling, whereas in the non-metastatic lines (WM3682, WM3526) Notch signaling is not active without stimulation [15]. First, we examined the effect of Notch signaling on MITF expression in the WM3682 cells, which express high levels of MITF [28] but lack endogenous Notch signaling [15]. Upon introduction of NICD, which mimics Notch signaling activation, MITF protein levels decreased in WM3682 melanoma cells (Figure 1A). Further, NICD expression in WM3682 and WM3526 melanoma cells led to a decrease in MITF pre-mRNA, as well as in its mature (spliced) mRNA form (Figure 1B). This suggests that Notch signaling inhibits MITF transcription. The positive control, Hes5, was upregulated upon Notch signaling activation. 

Sequence analysis of the MITF promoter revealed a potential conserved RBPJK binding site [33] in human (5′-TTCCAC-3′) and mouse (5′-TGAGAAA-3′ and 5′-CACTGTG-3′) (Figure 1C). To examine whether Notch signaling directly regulates MITF expression, we established a system in which Notch signaling is activated by external interaction with a Notch ligand that mimics physiological Notch signaling activation [15]. In this assay, Chinese hamster ovary) CHO) cells, which express Delta-like ligand 1 (DLL1) under the control of a doxycycline-inducible promoter, served as the sender cells [34]. The receiver cells were WM3682 melanoma cells transfected with a plasmid encoding a luciferase reporter gene driven by the MITF promoter (Figure 1D, left panel). Upon co-culturing these cells, Notch signaling activation reduced MITF promoter luciferase activity in the melanoma cells (Figure 1D, right panel). Finally, we evaluated MITF expression in WM3682 melanoma cells cultured on DLL1-coated plates with and without the γ-secretase inhibitor N-[N-(3,5-Difluorophenacetyl)-L-alanyl]-S-phenylglycine t-butyl ester (DAPT), which inhibits Notch signaling (Figure 1E, left panel). The reduction in MITF transcript levels due to culture on DLL1 was rescued upon Notch signaling repression (Figure 1E, right panel). These results suggest that Notch signaling inhibits MITF expression.

### 2.2. MITF Directly Regulates RBPJK Expression

We previously reported that MITF and RBPJK have co-evolved [32], and that RBPJK is a MITF co-factor necessary for induction of MITF transcriptional activity [15,32]. Conversely, we showed that Notch signaling decreases MITF expression (Figure 1). To gain better insight into the reciprocal interaction between Notch signaling and MITF levels, we examined the effect of MITF on RBPJK expression. Analysis of the RBPJK promoter revealed two conserved MITF binding sequences, known as E-box elements (5′-CACGCG-3′, Figure 2A). Further, MITF over-expression in melanoma cells WM3314 and WM1716, which normally express low levels of MITF [15], led to an increase in RBPJK mRNA levels (Figure 2B). MITF depletion by siMITF caused a reduction in RBPJK mRNA levels in WM3682 cells, which typically express high levels of MITF (Figure 2B). MITF over-expression in WM3314 melanoma cells, which express low levels of MITF, resulted in increased RBPJK protein levels (Figure 2C). To confirm that MITF occupies the RBPJK promoter, we employed a chromatin immunoprecipitation analysis to monitor markers of chromatin activity in WM3682 melanoma cells before and after MITF depletion by siMITF. We found that MITF reduction was accompanied by a decrease in histone 3 trimethylation at lysine 4 (H3K4me3) over the RBPJK promoter (Figure 2D). Since trimethylation is an epigenetic marker of transcriptionally active chromatin [35], these observations lend further support to the premise that MITF activates RBPJK transcription.

Notch signaling and MITF signaling impose opposite effects on melanoma invasive capacities: Whereas activation of Notch signaling enhances melanoma invasiveness [11,13,15] (Appendix A), MITF expression inhibits melanoma’s invasive capacity and promotes proliferation [19,23,24,25,26]. RBPJK, a key transcription regulator in the Notch signaling pathway [18], directly interacts with and mediates MITF transcriptional activity [15,32]. Notch signaling activation represses MITF activity in a competition-based mechanism by removing the RBPJK-MITF complex from MITF target gene promoters, leading to a melanoma invasive phenotype [15]. MITF, in turn, increases the expression of Notch target genes, such as Hes5 [15,18]. Here, we report an additional layer of reciprocal regulation between Notch and MITF signaling. In this transcription-based negative regulatory loop, Notch signaling represses MITF levels, and loss of MITF reduces the expression of RBPJK (Figure 3). These observations suggest that the Notch signaling-mediated negative feedback loop interferes with MITF signaling to maintain a melanoma invasive phenotype.

## 3. Materials and Methods

### 3.1. Cell Culture

WM3314, WM3526, and WM3682 melanoma cell lines were generously provided by Dr. Levi A. Garraway (Department of Medical Oncology and Center for Cancer Genome Discovery, Dana-Farber Cancer Institute, Boston, MA, USA). Cells were grown in a complete Dulbecco’s modification of Eagle medium (DMEM) medium supplemented with 10% fetal bovine serum (FBS, Sigma Aldrich, Rehovot, Israel) and 1% penicillin/streptomycin/glutamine (Invitrogen, Carlsbad, CA, USA). A PCR detection kit (Sigma-Aldrich, Rehovot, Israel) was used to test cells for mycoplasma. CHO-K1 cells stably expressing DLL1 under the control of the doxycycline-inducible promoter were kindly provided by Prof. David Sprinzak (Biochemistry Department, The George S. Wise Faculty of Life Sciences, Tel Aviv University, Tel Aviv, Israel). Plates coated with DLL1 [33] were prepared at a final concentration of 2 ng/μL DLL1. DAPT (Sigma-Aldrich, Rehovot, Israel) was added to the culture medium at a final concentration of 10 mM 48 h before analysis.

### 3.2. RNA Purification and qRT-PCR

Total RNA was extracted by Trizol (Invitrogen, Carlsbad, CA, USA) according to the manufacturer’s instructions followed by treatment with RNase-free DNase (QIAGEN, Hilden, Germany). RNA was quantified by determining the absorbance at 260 nm relative to 280 nm and was then subjected to one-step qRT-PCR using a MultiScribe RT-PCR kit (Applied Biosystems, Foster, CA, USA) and FastStart Universal SYBR Green Master (Roche, Basel, Switzerland) Data are presented as fold changes relative to control. All experiments were performed at least in triplicate. The standard error of the mean (SEM) is given. The PCR primers for the human genes are as follows: mature m-MITF forward, 5′-CATTGTTATGCTGGAAATGCTAGA-3′; mature m-MITF reverse, 5′-GGCTTGCTGTATGTGGTACTTGG-3′; Hes5 forward, 5′-AACTCCAAGCTGGAGAAGG-3′; Hes5 reverse, 5′-CTTCGCTGTAGTCCTGGTG-3′; RBPJK forward, 5′-GGCCTCCACCTAAACGACTTA-3′; RBPJK reverse, 5′-GCATGAAGAATAAGTACTGTT-3′; β-actin forward, 5′-ATTGCCGACAGGATGCAGAA-3′; β-actin reverse, 5′-GCTGATCCACATCTGCTGGAA-3′.

### 3.3. Melanoma Co-Culture with CHO-K1 Cells

WM3682 melanoma cells that express a green fluorescent protein (GFP) were transfected with the luciferase reporter gene plasmid. At 48 h after transfection, cells were co-cultured for 24 h with or without 10 mM DAPT (Sigma-Aldrich, Rehovot, Israel) at a 1:5 ratio to CHO-K1 cell numbers.

### 3.4. Plasmids and Transfection

The pcDNA3-MITF-HA expression vector was a gift from Dr. David. E. Fisher (Harvard Medical School, Boston, MA, USA). The pcDNA3-NICD expression vector and the Notch responsive element reporter gene were kindly provided by Dr. David Sprinzak (Tel Aviv University, Tel Aviv, Israel). For over-expression experiments, cells were transfected using the jetPEI kit (Polyplus-transfection, New York, NY, USA) according to the manufacturer’s instructions. For oligonucleotide transfection, siMITF or siControl were transfected into melanoma cells using HiPerFect (QIAGEN, Hilden, Germany) according to the manufacturer’s protocols. Cells were transfected twice with 100 pmol of siRNA per well (0.5 × 10^6^ cells) at 24-h intervals. Transfected cells were assayed 48 h after the second transfection. The sequences of the siRNAs are as follows: siControl, sense 5′-AAUUCUCCGAACGUGUCACGU-3′ and antisense 5′-ACGUGACACGUUCGGAGAAUU-3′; siMITF #1, sense 5′-GGCUUUCUAGAAAGAAUAA-3′ and antisense 5′-UUAUUCUUUCUAGAAAGCC-3′; siMITF #2, sense 5′-GGUGAAUCGGAUCAUCAAG-3′ and antisense 5′-CUUGAUGAUCCGAUUCACC-3′.

### 3.5. Luciferase Reporter Assay

For the reporter assays, 1 μg of reporter plasmid (MITF promoter-driven luciferase) was co-transfected into cells with 40 ng of the Renilla control plasmid using jetPEI (Polyplus-transfection, New York, NY, USA). Luciferase assays were performed using the Dual Luciferase Kit (Promega, Madison, WI, USA) according to the manufacturer’s recommendations. 

### 3.6. Western Blot Analyses

Proteins (40 μg) were resolved on 10% sodium dodecyl sulphate-polyacrylamide gel electrophoresis (SDS-PAGE) gels, followed by their transfer to nitrocellulose membranes (Whatman, Maidstone, UK). The membranes were incubated with antibodies to MITF (monoclonal C5; kindly provided by David Fisher, Harvard Medical School, Boston, MA, USA), β-actin (12620, Cell Signaling Technology, Danvers, MA, USA), RBPJK (sc-271128, Santa Cruz Biotechnology, Dallas, TX, USA), and cleaved Notch1 (Val1744, 4147, Cell Signaling Technology, Danvers, MA, USA), followed by incubation with appropriate horseradish peroxidase-conjugated antibodies. Proteins were detected by an enhanced chemiluminescence solution (Thermo Fisher Scientific, Waltham, MA, USA).

### 3.7. Chromatin Immunoprecipitation

Chromatin immunoprecipitation (ChIP) assays were performed, as described previously [15,32], with the MITF rabbit polyclonal antibody kindly provided by Dr. David Fisher (Harvard Medical School, Boston, MA, USA), rabbit anti-H3K4me3 (9725, Cell Signaling, Danvers, MA, USA), anti-H3K27me3 (9756, Cell Signaling, Danvers, MA, USA), and normal rabbit Immunoglobulin G IgG (sc-2027; Santa Cruz Biotechnology, Dallas, TX, USA) as a control. The PCR primers were as follows: RBPJK E-box 1, 5′-AGCTGGTCTAGGCAAACAC-3′ and 5′-GTTCTCGCGAGGTTTAGGAA-3′

### 3.8. Invasion Assays

WM3314 melanoma cells were treated with 20 µM PF3084014 [36] for 48 h and then seeded in duplicate in serum-free DMEM on an 8 µm pore Transwell membrane (Corning^®^, Corning, NY, USA) coated with Matrigel (BD Biosciences). Invasion analysis was conducted as previously described [15]. Images were analyzed using an IX81 microscope (Olympus, Tokyo, Japan) and cellSens Dimension software (https://www.olympus-lifescience.com/en/software/cellsens/). The number of cells that had invaded was normalized to the number of t otal 4′,6-diamidino-2-phenylindole (DAPI)-stained seeded cells.

## 4. Conclusions

Two modes of regulation are involved in the crosstalk between the Notch and MITF pathways in melanoma. Competition-based regulation: In the absence of cells that express the Notch ligand in the melanoma microenvironment, RBPJK directly binds to MITF and enhances MITF-mediated transcriptional upregulation of its target genes [32]. Upon activation of Notch signaling, NICD competes with MITF for binding to RBPJK, leading to the removal of the RBPJK-MITF complex from MITF’s target promoters [15]. Here we present a transcription-based regulation: MITF binds to the RBPJK promoter and positively regulates its expression. On the other hand, Notch signaling activation inhibits the transcription of the MITF gene and thus decreases MITF levels, resulting in an invasive and metastatic phenotype.

## Figures and Tables

**Figure 1 ijms-20-00576-f001:**
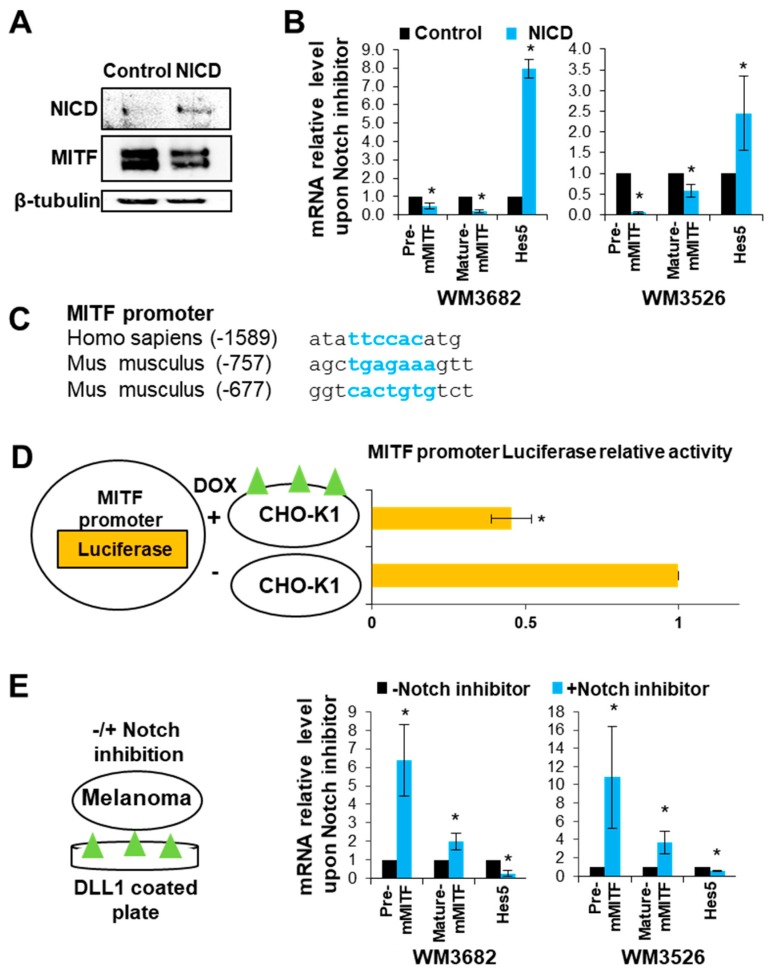
Notch signaling decreases microphthalmia-associated transcription factor (MITF) expression. (**A**) WM3682 cells were transfected with either Notch intracellular domain (NICD) cDNA or an empty plasmid. Western blot analysis was performed to determine MITF protein levels. β-tubulin was used as a loading control. (**B**) WM3526 and WM3682 cells were transfected with either NICD cDNA or an empty plasmid. qRT-PCR was performed to determine the levels of MITF pre-mRNA, mature MITF, and Hes5. Data were normalized to actin. Error bars represent the standard error of the mean (SEM), * *p* < 0.05 (*n* = 3). (**C**) Recombination signal binding protein J kappa (RBPJK) DNA binding sites (represent in blue) in the MITF promoter sequence. (**D**) Left panel: Experimental design scheme. Right panel: WM3682 cells were transfected with a MITF promoter reporter and then co-cultured with CHO cells stably transfected with a doxycycline-inducible DLL1-expression plasmid. DLL1 represents as green triangles. Firefly luciferase activity was normalized to Renilla luciferase activity. Error bars represent SEM, * *p* < 0.05 (*n* = 3). (**E**) Left panel: Experimental design scheme. Right panel: WM3526 or WM3682 cells were seeded on DLL1-coated plates and then treated with N-[N-(3,5-Difluorophenacetyl)-L-alanyl]-S-phenylglycine t-butyl ester (DAPT) (Notch inhibitor) or vehicle control dimethyl sulfoxide (DMSO). qRT-PCR was performed to determine the levels of MITF pre-mRNA, mature MITF mRNA, and Hes5. Data were normalized to actin. Error bars represent SEM, * *p* < 0.05 (*n* = 3).

**Figure 2 ijms-20-00576-f002:**
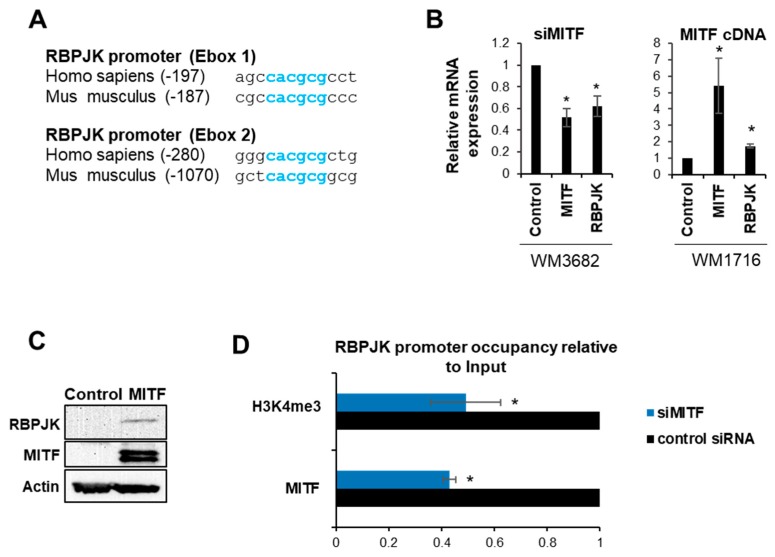
RBPJK increases MITF expression. (**A**) Two conserved MITF DNA binding sites (E-boxes, represent in blue) in the RBPJK promoter sequence. (**B**) Melanoma cells with high levels of MITF (WM3682, left panel) and low levels of MITF (WM1716, right panel) were treated with siMITF or MITF cDNA, respectively, followed by RBPJK expression level analysis. As controls, cells were treated with siControl or empty cDNA, respectively. Expression levels were normalized to Glyceraldehyde 3-phosphate dehydrogenase (GAPDH). Error bars represent SEM, * *p* < 0.05 (*n* = 3). (**C**) Western blot analysis of RBPJK and MITF protein levels in cells transfected with MITF cDNA or an empty plasmid. Actin was used as loading control. (**D**) Chromatin immunoprecipitation (ChIP) was performed on extracts from WM3682 cells transfected with siMITF or siControl. Protein:chromatin-crosslinked complexes were precipitated with the indicated antibodies. PCR primers spanning the region encoding the RBPJK promoter were used. The data show promoter occupancy relative to input. Error bars represent SEM, * *p* < 0.05 (*n* = 3).

**Figure 3 ijms-20-00576-f003:**
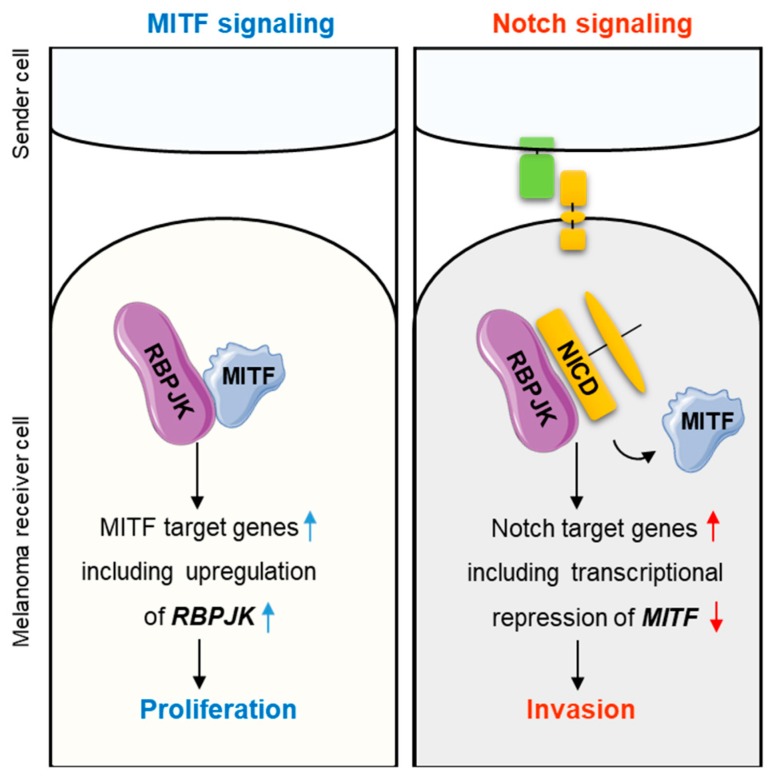
Two modes of regulation are involved in the crosstalk between the Notch and MITF pathways in melanoma. Competition-based regulation: In the absence of cells that express the Notch ligand in the melanoma microenvironment, RBPJK directly binds to MITF and enhances MITF-mediated transcriptional upregulation of its target genes. Upon activation of Notch signaling, NICD competes with MITF for binding to RBPJK, leading to the removal of the RBPJK-MITF complex from MITF’s target promoters (Golan 2015). Transcription-based regulation: MITF binds to the RBPJK promoter and positively regulates its expression. Notch signaling activation inhibits the transcription of the MITF gene and thus decreases MITF levels, resulting in an invasive and metastatic phenotype. Blue arrow represents MITF target genes; Red arrow represents Notch signaling target genes; Green Shape represents DLL1; Yellow shape represents Notch receptor (intact or cleaved). Blue letters represent induction of MITF signaling and the proliferative phenotype; Red letters represent induction of Notch signaling and the invasion phenotype. 3. Materials and Methods.

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
