# Peer review of "Negative Regulatory Loop between Microphthalmia-Associated Transcription Factor (MITF) and Notch Signaling"

_ijms, 2019, doi:10.3390/ijms20030576_

Reviewer 1 Report

The paper discussed a negative regulatory loop between MITF and Notch signaling could suppress MITF expression and maintain melanoma cells invasiveness and metastatic phenotype, however, the paper lack of in depth study and discussion of how these two pathway are interacted to make this conclusion. 

1) First, in the paper, the author cited their publication in 2015, which seems the similar finding has been reported before? 

2) Several paper showed that MITF contribute to the melanoma metastasis which seems to be  controversial to the findings in this paper. Does author has any explanation?

3) In the Figure 1a, the melanoma cells were transfected with NICD cDNA, does not melanoma cells express NICD by itself? what is the endogenous level of  NICD expression in the two cell lines used in the paper? And Fig 1A the western of NICD is not very clear. 

4) A transwell assay or cell migration assay should be include at least to support the claim as in the conclusion that activated Notch signaling repress MITF signaling and promote the melanoma metastasis.

Author Response

Reviewer 1

The paper discussed a negative regulatory loop between MITF and Notch signaling could suppress MITF expression and maintain melanoma cells invasiveness and metastatic phenotype, however, the paper lack of in depth study and discussion of how these two pathway are interacted to make this conclusion. 

1) First, in the paper, the author cited their publication in 2015, which seems the similar finding has been reported before? 

We thank the reviewer for the insightful comments. We believe that addressing the issues raised improved our manuscript. In our 2015 paper, "Interactions of Melanoma Cells with Distal Keratinocytes Trigger Metastasis via Notch Signaling Inhibition of MITF", we demonstrated that interaction of melanoma cells with differentiated keratinocytes activates Notch signaling in melanoma cells. We found that the Notch intracellular domain (NICD) inhibits MITF function by upregulating the expression of miR-222/221 to trigger the critical transit into the invasive stage. We also demonstrated the molecular mechanism, by which RBPJK (a Notch central transcription mediator) directly interacts with and mediates the repressive ability of MITF. The NICD competes with MITF for binding to RBPJK regulating the interaction of MITF with target promoters. Thus, our previous work demonstrated a competition-based mechanism, whereas in the current manuscript we report on an additional layer of transcription-based regulation that exists between Notch and MITF pathways. In the manuscript under review, we showed that induction of Notch signaling activation inhibits the transcription of the MITF gene and thus decreases MITF levels. In addition, we found that MITF increases RBPJK expression by directly binding to the RBPJK promoter. In order to emphasize the novelty of our results, we have modified the abstract, the text, and the graphical abstract.

2) Several paper showed that MITF contribute to the melanoma metastasis which seems to be controversial to the findings in this paper. Does author has any explanation?

To address this important comment, we have modified the relevant paragraph in the Introduction section. Melanoma progression is currently thought to occur via a "phenotypic switch" model by which cells continuously switch between proliferative and invasive phenotypes. In the early stage, cells express a proliferative signature gene set that allows proliferation at the primary lesion. As disease progresses, microenvironmental cues trigger expression of the invasive gene set. These cells are able to migrate out of the primary site to a suitable distal site. Once at the metastatic location, the melanoma cells revert to the proliferative state, and a new cycle is started (Hoek 2008). Therefore, for the generation of metastases two mutually exclusive processes are crucial: proliferation and invasion. MITF is the key regulator of the melanoma phenotypic switch; this one factor transcriptionally regulates the proliferative and the invasive sets of gene. Cells that express high levels of MITF exist in a proliferative and weakly invasive state, whereas cells with low MITF levels are highly invasive and less proliferative (Hoek 2006, Hoek 2008, Carreira 2006). Therefore, although it might seem controversial, MITF contributes to melanoma metastasis by driving cell proliferation, while in parallel blocking invasive potential.

 3) In the Figure 1a, the melanoma cells were transfected with NICD cDNA, does not melanoma cells express NICD by itself? what is the endogenous level of NICD expression in the two cell lines used in the paper?

The levels of the receptors in the Notch family in various melanoma cell lines were previously determined by us (Golan et al 2015). Highly invasive melanoma cell lines, which express low level of MITF, express higher levels of Notch receptors than do melanoma lines that express high levels of MITF. Additionally, we showed that there is no endogenous Notch signaling in WM3682 cells. Therefore, these data were not presented in the current manuscript. To clarify this important point, we have now added the relevant text and references to the manuscript.

Fig 1A the western of NICD is not very clear. 

We have replaced Figure 1A with a better image.

4) A transwell assay or cell migration assay should be include at least to support the claim as in the conclusion that activated Notch signaling repress MITF signaling and promote the melanoma metastasis.

We performed the suggested transwell invasion assay. This analysis demonstrated the inhibitory effect of the Notch inhibitor PF3084014 (Lanza 2015) on the invasion capacity of WM3314 melanoma cells (new Supplementary Figure S1). We have modified the main text to describe this result and have added a new section to the Methods to describe the assay.

Reviewer 2 Report

It is an interesting short study based on previous results.

My question is that are the data/methods applicable for melanoma samples?

Author Response

It is an interesting short study based on previous results.

My question is that are the data/methods applicable for melanoma samples?

We thank the reviewer for his insightful question. Notch signaling activation requires direct contact with the Notch ligand, as we demonstrated in two systems (Sprinzak et al., 2010): a co-culture of melanoma cells with Notch ligand expressing cells (Described in Figure 1D) and a culture of melanoma cells on DLL1-coated plates (described in Figure 1E).  We are confident that these methods will apply to melanoma samples from patients as well as other types of tumor samples.  We will be happy to share our knowledge and reagents to allow other labs to use these methods.

Reviewer 3 Report

  This is a brief, clear and smart study demonstrating the existence of a negative regulatory loop between Notch signaling and MITF. Introduction is focused on the right two poles, Notch activation and mechanism, as well as MITF roles in the survival, proliferation and suppression of metastasis capacity in melanoma cells.

 The manuscript is well referenced, but it seems to me that something is wrong concerning reference numbering.  At line 63, authors begins saying “we….. “ , using ref. 13, but the related reference of this lab is 15. Reference 28 is related to uveal melanoma, and it does not fit with the use of ref. 28 throughout the manuscript. Other numbers seem to be correct. So, please, check and verify reference numbering.

Figure 1, panel B, referred to WM3526 cells. Decimal figures should be added to the y-axis.

Line 117: Figure 2E is mentioned. Where is it?  There is no Figure 2E.

 Minor points, concerning Methods:

Is DAPT N-[N-(3, 5-difluorophenacetyl)-L-alanyl]-S-phenylglycine? If so, it would be helpful for readers an abbreviation list for this one and other abbreviations.

Some reagents (plasmid and antibodies) are provided by Dr. Fischer, and the uncompleted/ completed address of Dr. Fischer is written every time. Homogenize the address and once is enough.

The annealing positions of the primers used in paragraph 3.2 of the method section would also be helpful.

Author Response

Reviewer 3

This is a brief, clear and smart study demonstrating the existence of a negative regulatory loop between Notch signaling and MITF. Introduction is focused on the right two poles, Notch activation and mechanism, as well as MITF roles in the survival, proliferation and suppression of metastasis capacity in melanoma cells.

The manuscript is well referenced, but it seems to me that something is wrong concerning reference numbering.  At line 63, authors begins saying “we… “, using ref. 13, but the related reference of this lab is 15. Reference 28 is related to uveal melanoma, and it does not fit with the use of ref. 28 throughout the manuscript. Other numbers seem to be correct. So, please, check and verify reference numbering.

We apologize for this error. We have corrected the references mentioned and verified that the rest of the references are appropriately cited.

Figure 1, panel B, referred to WM3526 cells. Decimal figures should be added to the y-axis.

We have added a decimal figure to the y axis of Figure 1B.

Line 117: Figure 2E is mentioned. Where is it?  There is no Figure 2E.

We have corrected this error.

Minor points, concerning Methods:

Is DAPT N-[N-(3, 5-difluorophenacetyl)-L-alanyl]-S-phenylglycine? If so, it would be helpful for readers an abbreviation list for this one and other abbreviations.

We now give the complete names for DAPT and other reagents.

Some reagents (plasmid and antibodies) are provided by Dr. Fischer, and the uncompleted/ completed address of Dr. Fischer is written every time. Homogenize the address and once is enough.

We now give the full name and institutional address for Dr. Fischer only once.

The annealing positions of the primers used in paragraph 3.2 of the method section would also be helpful.

We have added the annealing positions of the primers.